# The Importance of Mitral Valve Prolapse Doming Volume in the Assessment of Left Ventricular Stroke Volume with Cardiac MRI

**DOI:** 10.3390/medsci11010013

**Published:** 2023-01-24

**Authors:** Rui Li, Hosamadin Assadi, Gareth Matthews, Zia Mehmood, Ciaran Grafton-Clarke, Bahman Kasmai, David Hewson, Richard Greenwood, Hilmar Spohr, Liang Zhong, Xiaodan Zhao, Chris Sawh, Rudolf Duehmke, Vassilios S. Vassiliou, Faye Nelthorpe, David Ashman, John Curtin, Gurung-Koney Yashoda, Rob J. Van der Geest, Samer Alabed, Andrew J. Swift, Marina Hughes, Pankaj Garg

**Affiliations:** 1Norwich Medical School, University of East Anglia, Norfolk NR4 7TJ, UK; 2Norfolk and Norwich University Hospitals NHS Foundation Trust, Norfolk NR4 7UY, UK; 3National Heart Research Institute Singapore, National Heart Centre Singapore, 5 Hospital Drive, Singapore 169609, Singapore; 4Cardiovascular Sciences Academic Clinical Programme, Duke-NUS Medical School, 8 College Road, Singapore 169856, Singapore; 5Cardiology Department, Queen Elizabeth Hospital King’s Lynn NHS Foundation Trust, King’s Lynn PE30 4ET, UK; 6Department of Radiology, Leiden University Medical Center, 2333 ZA Leiden, The Netherlands; 7Department of Infection, Immunity and Cardiovascular Disease, University of Sheffield, Sheffield S10 2TN, UK; 8Department of Clinical Radiology, Sheffield Teaching Hospitals NHS Foundation Trust, Sheffield S10 2JF, UK

**Keywords:** mitral valve prolapse, AI, reliability, flow quantification

## Abstract

There remains a debate whether the ventricular volume within prolapsing mitral valve (MV) leaflets should be included in the left ventricular (LV) end-systolic volume, and therefore factored in LV stroke volume (SV), in cardiac magnetic resonance (CMR) assessments. This study aims to compare LV volumes during end-systolic phases, with and without the inclusion of the volume of blood on the left atrial aspect of the atrioventricular groove but still within the MV prolapsing leaflets, against the reference LV SV by four-dimensional flow (4DF). A total of 15 patients with MV prolapse (MVP) were retrospectively enrolled in this study. We compared LV SV with (LV SV_MVP_) and without (LV SV_standard_) MVP left ventricular doming volume, using 4D flow (LV SV_4DF_) as the reference value. Significant differences were observed when comparing LV SV_standard_ and LV SV_MVP_ (*p* < 0.001), and between LV SV_standard_ and LV SV_4DF_ (*p* = 0.02). The Intraclass Correlation Coefficient (ICC) test demonstrated good repeatability between LV SV_MVP_ and LV SV_4DF_ (ICC = 0.86, *p* < 0.001) but only moderate repeatability between LV SV_standard_ and LV SV_4DF_ (ICC = 0.75, *p* < 0.01). Calculating LV SV by including the MVP left ventricular doming volume has a higher consistency with LV SV derived from the 4DF assessment. In conclusion, LV SV short-axis cine assessment incorporating MVP dooming volume can significantly improve the precision of LV SV assessment compared to the reference 4DF method. Hence, in cases with bi-leaflet MVP, we recommend factoring in MVP dooming into the left ventricular end-systolic volume to improve the accuracy and precision of quantifying mitral regurgitation.

## 1. Introduction

Valvular heart disease prevalence is expected to increase worldwide due to population ageing [1]. Mitral regurgitation (MR) is the second most common type of valvular heart disease in Europe [2], accounting for a quarter of cases. Mitral valve prolapse (MVP), particularly secondary to myxomatous degeneration, is the leading cause of primary non-ischaemic mitral regurgitation (MR). MR and MVP are associated with significant morbidity and mortality [3,4], especially in the female population [5,6], in which long-term severe ventricular arrhythmia present notable excess mortality and mortality [7].

Mitral regurgitation assessment is mainly conducted by echocardiography. An integrated approach of several parameters is the preferred approach advocated by the American College of Cardiology/American Heart Association Guidelines for the management of patients with valvular heart disease and also by the European Society of Cardiology/European Association of Cardiothoracic Surgery Guidelines [8,9]. These parameters include vena contracta, proximal iso velocity surface area (PISA), MR regurgitation volume, and effective regurgitant orifice area. However, previous studies have questioned the reliability and the interobserver agreement of vena contracta and PISA techniques for MR quantification [10]. Moreover, previous work by Uretsky et al. demonstrates that there is significant discordance between these echocardiographic parameters for the grading of mitral regurgitation [11]. Importantly, cases with bileaflet MVP are more likely to have multiple mitral regurgitation jets where these surrogate parameters of MR quantification become impractical, and their clinical value is debatable. MR jet eccentricity can also make it challenging to quantify mitral regurgitation by echocardiography. It is important to note that all these parameters can be difficult to image, especially in patients with poor echocardiographic acoustic windows. Hence, even though echocardiography remains the main imaging method, complimentary imaging methods are needed for further assessment of mitral regurgitation for optimum clinical decision-making regarding intervention.

Cardiac magnetic resonance (CMR) imaging is the gold standard non-invasive modality for biventricular volume quantification [12,13] and preserves advantages to characterise mitral annulus disjunction and myocardial fibrosis which are associated with arrhythmogenesis [14,15]. Volumetric measurement techniques for MR quantification, which incorporate left ventricular (LV) stroke volume (SV), are considered to be the most accurate quantification methods [16]. The standard method of LV SV measurement (LV SV_standard_), obtained by subtracting the aortic phase-contrast forward volume (AoPC) from the left ventricular (LV) stroke volume (SV), is the most widely researched CMR assessment [17]. LV SV is quantified by subtracting the left ventricular end-systolic volume (LVESV) from the left ventricular end-diastolic volume (LVEDV). Imprecise measurement of LVESV, and therefore LV SV, results in an inaccurate MR grading.

In patients with bi-leaflet mitral valve prolapse, LV SV assessment is challenging due to the significant doming of the mitral valve leaflets on the atrial side of the atrioventricular groove. It remains debated whether the blood volume trapped on the ventricular side of the doming leaflets during systole should be included in the left ventricular end-systolic volume and whether this will impact the quantification of MR.

Four-dimensional flow (4DF) CMR allows quantifying transvalvular flow precisely using valve tracking procedures to generate a true reformatted plane through most mobile valves—mitral and tricuspid. Both mitral and aortic flows can quantify LV SV, applying the conservation of mass principle. Because of the technical advantage, 4DF CMR has become the new gold standard for flow quantification [18].

This study aims to test the difference between the measurement of LV SV with and without the MVP left ventricular doming volume, compared with the reference method of 4DF for the quantification of LV SV.

## 2. Materials and Methods

### 2.1. Study Cohort

Fifteen cases with MVP from our routine CMR service from February 2021 to March 2022 were retrospectively enrolled in this study. All were adult patients, who were clinically stable as outpatients, and had baseline functional cine images and 4DF CMR assessment data. Exclusion criteria were limited to any CMR contraindication (e.g., pacemaker, defibrillator). No patient was consecutively enrolled.

### 2.2. CMR Protocol LV Volume Assessment

CMR study was conducted on a 3 Tesla Discovery MR750w GE system (GE Healthcare, Milwaukee, WI, USA). The CMR protocol included baseline survey images and cines with 30 phases. Cine images were acquired during end-expiratory breath-hold with a Gated 2D FIESTA (Fast Imaging Employing Steady-state Acquisition) single-slice breath-hold sequence. Long-axis Gated 2D FIESTA cine in four-chamber, three-chamber, and two-chamber planes and short-axis Gated 2D FIESTA cines were also acquired. The number of left ventricular (LV) short-axis slices depended on each patient’s heart size. Cine imaging, gadolinium enhancement imaging, and 4DF acquisition methods have been previously published by our group [19,20,21].

LV volumes were quantified in a short-axis (SAX) stack using CVI42 version 5.14 (Circle Cardiovascular Imaging Inc., Calgary, Canada). End-diastolic and end-systolic phases were manually defined, and contours were drawn automatically using artificial intelligence (AI), and then visually checked by an experienced operator. An expert with more than 10-year CMR analysis experience supervised and checked the quality of manual contour and independently interpreted the results.

Papillary muscles were manually excluded from the LV volume. LVESV, LVEDV, and LV SV were recorded. In the first group, LV SV_standard_ was obtained (LV SV_standard_ = LVEDV – LVESV) thus excluding MVP left ventricular doming volume. In the second group, MVP left ventricular doming volume was included in LV SV assessment (LV SV_MVP_) by manually drawing contours where AI failed to factor in MVP left ventricular doming volume (Figure 1c,d).

### 2.3. 4DF CMR Acquisition

The initial VENC setting for 4DF CMR was 150–200 cm/s for all cases. This was optimised depending on beforehand available echocardiography data. 4DF CMR acquisition prescribed k-t adaptive Accelerated Cartesian MRI, namely kat-ARC or Hyperkat, a spatiotemporal-correlation-based autocalibrating parallel imaging method with cardiac motion adaptive temporal window selection [22]. The k-t sampling scheme used adaptable density to improve accuracy and reduce coherent residual artefacts. In addition, a static tissue removal scheme was adopted to identify voxels with limited flow or motion and delete the signal from such static voxels prior to Hyperkat processing. This decreases residual aliasing artefacts at their high acceleration during the reconstruction. Field-of-view of the acquisition was planned to cover the whole heart, aortic valve, and proximal ascending aortic root only. HyperKat acceleration with a factor of 6 was used. Other standard scan parameters were, field-of-view = 340 mm × 340 mm, acquired voxel size = 3 × 3 × 3 mm^3^ and reconstructed voxel size = 1.5 × 1.5 × 1.5 mm^3^. Flip-angle was 8°, with TE (ms) of 2.14 and TR (ms) of 4. Electrocardiogram gating was retrospective to avoid diastolic temporal blurring. Respiratory compensation was free-breathing. The acquired temporal resolution was 40 ms. The number of phases was kept consistent to cines at 30 cardiac phases.

### 2.4. 4D Flow CMR Analysis

4DF analysis through the mitral valve and aortic valve was performed using CAAS MR Solutions (Version 5.1, Pie Medical Imaging, Maastricht, Netherlands), with automated velocity offset correction applied. An expert with more than 10-year 4DF and CMR analysis experience supervised and checked the quality of manual contour and independently interpreted the results. Automated valve tracking was done for two orthogonal views of the mitral and aortic valves, with manual correction applied on the region of interest contours if appropriate (Figure 1e,f). Aortic backward flow (ABF) and mitral forward flow (MVF) were recorded to calculate LV SV (LV SV_4DF_) using the following equation of conservation of mass principle applied to blood flow haemodynamics: LV SV_4DF_ = ABF + MVF.

### 2.5. MR Severity Assessment

MV regurgitant fraction (RF) was calculated by the ratio between mitral backward flow volume derived by 4DF and the respective LVSV_standard_/LVSV_MVP_. An RF < 5% is defined as absent MR, 5–29% mild MR, 30–49% moderate MR, and ≥50% severe MR [23].

### 2.6. Statistical Analysis

Data analyses were performed using SPSS statistical package (version 28.0, IBM, Chicago, Illinois, USA). Continuous variables were recorded as mean ± standard deviation (SD). Shapiro–Wilk test was used to test the normality of parameters, followed by hypothesis testing with Student’s *t*-test or a Mann–Whitney U test as appropriate. Inter-observer correlation coefficient (ICC) estimates and associated 95% confidence intervals were calculated based on the absolute-agreement, 2-way mixed-effects model. Paired Student’s *t*-test was performed to compare LV SV_standard_ and LV SV_MVP_. A *p* value <0.05 is considered statistically significant.

## 3. Results

Patient characteristics: Patient demographics are summarised in Table 1. A total of 15 patients were included in this study, of which 11 (68%) were female. The mean age of our study population was 50 ± 20 years. The mean body surface area (BSA) was 1.91 ± 0.2 m^2^. All patients were in sinus rhythm, 14% had a history of hypertension and 33% were current or ex-smokers. More than half were in New York Heart Association (NYHA) class I (60%), four (27%) were in NYHA class II, two (13%) were in NYHA class III and one (7%) was in NYHA class IV. The most used long-term medications included beta-blockers (60%), angiotensin-converting enzyme inhibitors (40%), diuretics (27%), and calcium channel antagonists (7%).

Descriptive statistics for the recorded parameters are presented in Table 2. LV SV_standard_ was significantly greater than either LV SV_MVP_ and LV SV_4DF_ (Figure 2) with *p* < 0.001 and *p* = 0.02, respectively. No difference was observed between LV SV_MVP_ with LV SV_4DF_ (*p* = 0.6). The ejection fraction in the LV SV_MVP_ group was significantly lower than the LV SV_standard_ group (52% ± 11% vs. 61% ± 14%, *p* < 0.001).

MV ejection fraction was higher when calculated using mitral valve backward volume to LV SV_MVP_ with an average increase of 2.1% compared with using LV SV_standard_. The MR severity of none/mild/moderate/severe was slightly different when calculated using LV SV_MVP_ compared to LV SV_standard_, changed from 4/8/2/0 to 3/9/2/0, respectively. The MR ejection fraction increased by an average of 0.3%, 2.3%, and 4.3% in none, mild, and moderate MR groups, respectively.

ICC test was used to assess the agreement between the reference method and LV SV_standard_, and between the reference method and LV SV_MVP_. LV SV_4DF_ and LV SV_standard_ achieved a moderate ICC score of 0.75 (*p* < 0.01) but was better with LV SV_MVP_ where the agreement was good, ICC = 0.86 (*p* < 0.001).

## 4. Discussion

In this study, we highlight the importance of including the MPV left ventricular doming volume within the left ventricular end-systolic volume during routine clinical assessment by CMR. This volume is the LV volume which lies on the ventricular side of the mitral valve, but on the atrial aspect of the atrioventricular groove, during end-systole. The doming volume is not crossing through the mitral valve in systole and does not contribute to the regurgitant volume of the mitral valve. We demonstrated that by including the MVP doming volume, the LV SV shows better agreement with the reference 4DF measured LV SV. When the MVP doming volume is not included in the LV SV, this results in significantly higher LV SV compared to 4DF-derived LV SV, potentially resulting in over-estimation of the left ventricular ejection fraction. Our study demonstrated that including MVP doming volume within the left ventricular end-systolic volume is important for precisely quantifying LVSV in patients with MVP. If MVP doming volume is not accounted for, mitral regurgitation quantification may be overestimated, leading to inaccurate grading and potentially affecting clinical decision-making regarding timely intervention.

AI auto-contouring algorithms in most commercial cardiac post-processing software rely on the detection of mitral and aortic valves and the apex of the left ventricle for the left ventricular volume assessment, which is proven to provide significant improvement in accuracy and reproducibility [24]. However, for the MVP cases, in which the mitral valve prolapse doming volume during the end-systolic phase remains debated whether to be included or excluded into the left ventricular volume, AI segmentation automatically excludes the doming volume from the left ventricular volume as it lies above the AI detected mitral valve level. The complexity of the mitral valve detection in MVP cases hampers the AI performance of ventricular volume assessment. Researchers have recognised this limitation and are developing new solutions to incorporate complex valve disease cases. Jin et al. tested a novel AI technique called Anatomical Intelligence in ultrasound which semiautomatically tracks the annulus and leaflet anatomy for parametric analysis and concluded the novel AI technique provides superior accuracy compared to non-expert manual segmentation with significantly less time required for image analysis. However, it still underperforms compared to expert segmentation [25]. In complex ventricular valve cases, visual supervision and manual correction of the AI-derived contours by an expert are still needed to provide precise ventricular volume assessment results.

Transthoracic echocardiography (TTE) is recommended as a first-line imaging test for valvular heart disease assessment by the European Society of Cardiology guidelines. However, TTE typically has methodological limitations due to dependency on flow convergence region geometric assumptions and doppler measurement angle dependency. When various echocardiographic methods give inconsistent MR grading, CMR is recommended for further precise assessment of not only MR severity but also volume assessment [8]. Furthermore, late gadolinium enhancement and T1 mapping assessments for microfibrous using CMR were suggested to stratify MVP patients at risk for malignant arrhythmias, and may further contribute to the identification of different MVP phenotypes [26]. Overall, CMR has been reported to identify mitral valve prolapse with a sensitivity and specificity of 100% [27] and has become a worldwide routine cardiac assessment tool for follow-up of patients with MVP-related moderate to severe MR and surgical decision-making [28].

MR is reported to have a prevalence in 36.9% of adult patients with MVP and 35.7% of teenagers with MVP [29,30], where MR volume increased by more than 8 mL in 51% of MVP patients during a 1.5 year follow-up period [31]. These studies suggest MVP is a progressive disease leading to the occurrence and progression of MR, though the progression is small in magnitude overall and has more clinical relevance in patients with moderate or severe MR. The precise diagnosis of MVP and assessment of MR, especially for patients with mild and moderate MR, is crucial for an appropriate patient management to prolong the life expectancy in clinical settings.

The management of MVP remains a challenge due to the lack of standardised risk-stratification models. A recent study presented a mechanistic approach to sudden death prevention in mitral valve prolapse patients and classified MVP patients with a history of ventricular fibrillation and ventricular scarring as a high-risk group. Implantable cardioverter defibrillators should be strongly considered for high-risk patients to prevent sudden death and conservative management with cardiac surveillance monitoring with or without medical therapy is recommended for low-risk MVP patients [32]. Although mitral valve repair is the suggested treatment for MVP patients with severe mitral regurgitation, the surgery often leaves patients with LV dysfunction and progressive residual fibrosis [33]. However, long-term survival could be improved if the surgery is performed in an earlier disease process stage [34]. It still remains under-researched if earlier surgery could lead to less fibrosis and sudden death, more long-term data are needed to identify the correlation between MVP-induced fibrosis and long-term mortality.

4DF CMR offers a three-directional velocity-encoded dataset enabling quantification of peak velocities and transvalvular blood flow with improved precision. This is crucial in assessing complex valvulopathy. Previous studies looked at the clinical utility of 4DF CMR against routine CMR and TTE for MR quantification in MVP [35]. Spampinato et al. found a moderate to strong correlation and good to excellent intra- and interobserver variability between the three methods [35]. However, TTE overestimated MR volume when compared to 4DF flow CMR (mean difference of 17.2 mL, 95% CI 8.4–25.9, *p* < 0.001) [35]. Our study adds to their research by demonstrating that this MVP doming volume needs to be factored into the LVESV to further improve MR quantification in complex bi-leaflet MVP cases with significant dooming. This is likely to reduce the MR grading further by CMR when compared to TTE methods.

Other studies examined the impact of factoring in the MVP doming volume to assess LV function and MR severity by CMR assessment. When accounting for the MVP doming volume, 66% of patients with bi-leaflet mitral valve prolapse were recategorised, suggesting that without factoring in the MVP doming volume, one could underestimate MR severity by 1 grade in two-thirds of patients [36]. Another study examined the impact of prolapse doming volume on MR quantification in MVP patients and suggested the mitral regurgitant volume could be significantly underestimated in MVP patients with a prolapse doming volume greater than 14 mL [37]. Our study further evidenced that MVP volume should be counted into the LVESV in patients with MVP by exploring the consistency between LV SV factored in MVP volume with 4DF-derived LV SV. This is of utmost significance as moderate to severe MR are important public-health problems and have an increased risk of major adverse cardiac events and death [3,38]. Untreated severe MR confers up to 14% annual mortality risk [39]. An inaccurate MR severity grading could also lead to inappropriate clinical decision-making for the patients, as different therapeutic approaches and/or surgery are recommended in managing moderate and severe secondary MR. Accordingly, precise quantification of MR is crucial to avoid unnecessary operations. Early surgery in patients with mild or moderate MR could risk them having to re-do the surgery later in life and may increase morbidity. Moreover, an overestimation of MR severity may lead to additional unnecessary follow-ups and create anxiety for the patient.

Mitral annular disjunction (MAD) has a strong association with ventricular arrhythmias and sudden cardiac death [40], with a prevalence of 30.1% in MVP patients [41], and is independently associated with excess risk for arrhythmic events [42]. Previous studies have suggested that MVP strongly predicts MAD [43]. Although without consistent definition and its clinical implication remains under-researched, MAD is an increasingly recognised finding amongst patients undergoing cardiac imaging and has been reported in 23.1% of MVP and sudden cardiac death [44]. Its association with an increased risk of severe MR and prognosis requires further investigation. Therefore, CMR offers a one-stop complete assessment of MR severity, possible hints to the aetiology of MR, MAD, and volume overload assessment of MR on the LV.

In routine MVP cases, the quantification of MR by CMR needs to be precise. In this paper, we used 4DF MRI as a method for further optimisation of LVSV by the most standard methods of short-axis cine stack segmentation which is routinely used across the globe in clinical practice. The presently described method could be incorporated into clinical practice to improve the routine practice of left ventricle segmentation of short-axis cine stack and MR quantification, requiring prospective studies to ascertain if outcomes are improved.

## 5. Limitations

This is a single-centre retrospective observational study. Future studies are needed to confirm our findings in multi-centre prospective studies. The data set in this study is relatively small in scale, potentially leading to insufficient study power and type II error, although the effect size was large and standard deviations consistent. More extensive prospective studies are needed to explore the clinical impact of improved precision of left ventricular segmentation and its impact on clinical management. There is a mixed male and female population in this study however with the majority of female participants (73.3%), which is in keeping with the prevalence of MVP. The gender differences could be further explored where a balanced number of participants from each gender and a larger scale of data are available. Only two MVP patients (13.3%) in this study are classified as moderate MR and no severe MR patient with MVP was in our retrospective CMR registry, in which factoring in MVP doming volume could have a greater difference in calculating LV SV and thus MR severity could be recategorised with a higher percentage in moderate or severe MVP patients. Finally, no direct MR grading comparison with 4DF assessment was performed in this study, as the aim of this study was not to compare the performance of two MR grading methods but to highlight the importance of factoring in MVP doming volume into LVESV during routine CMR assessment.

## 6. Conclusions

This study shows that when LV SV short-axis cine assessment incorporates MVP dooming volume, it significantly improves the precision of LV SV assessment compared to the reference 4DF method. Hence in routine clinical CMR practice, in cases with bi-leaflet MVP, MVP dooming needs to be factored into the left ventricular end-systolic volume to improve the accuracy and precision of quantifying mitral regurgitation.

## Figures and Tables

**Figure 1 medsci-11-00013-f001:**
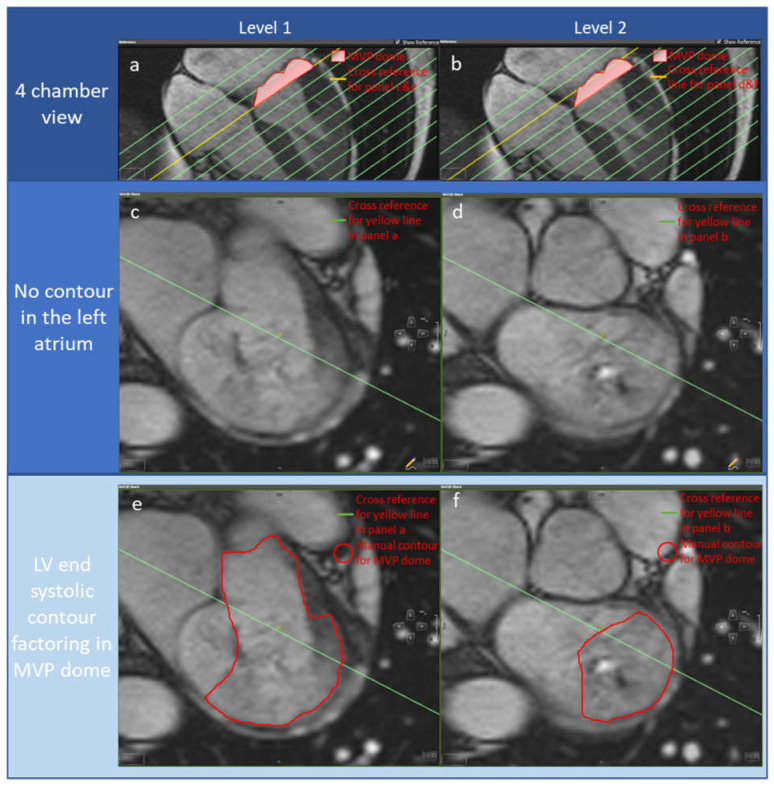
A case example of a mitral valve prolapse patient wherein AI failed to factor in the prolapse, resulting in a 34 mL difference at the systole phase. Panel (**a**,**b**): yellow reference lines for images in panels (**c**,**e**), and (**d**,**f**), respectively. Panel (**c**,**d**): SAX slides at the end-systole phase without including the MVP doming volume. Panel (**e**,**f**): SAX slides at the systole phase in which manual refinements (red contour) are applied to include the MVP doming volume in LVESV.

**Figure 2 medsci-11-00013-f002:**
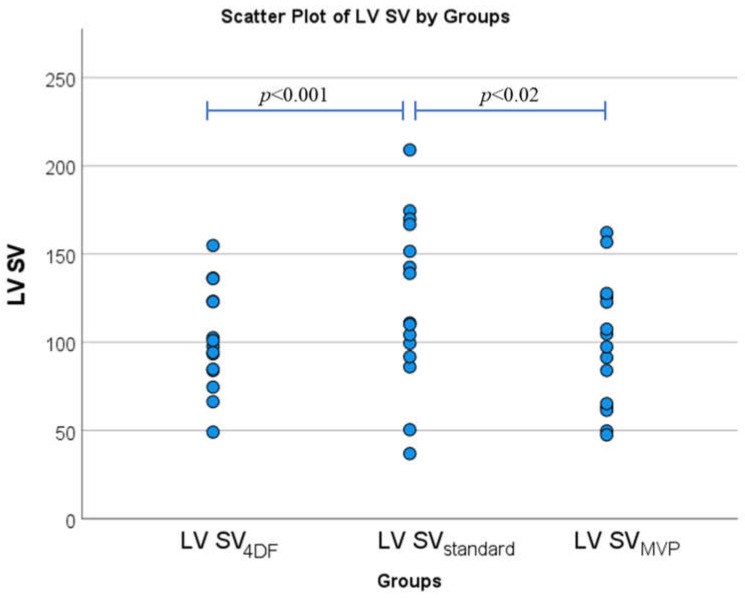
Bar Charts illustrating the comparison of mean LV SV assessment using paired *t*-tests between the three groups (LV SV_4DF_, LV SV_standard_, and LV SV_MVP_) (*n* = 15).

**Table 1 medsci-11-00013-t001:** Demographic variables of the 15 patients included in this study.

Baseline Characteristic	*n* (%) or Mean ± SD
Male	4 (26.7%)
Female	11 (73.3%)
Age (years)	49.8 ± 19.6
BSA (m^2^)	1.91 ± 0.2
Sinus rhythm	13 (86.7)
DM	0 (0%)
HTN	2 (13.3%)
Previous MI	0 (0%)
Smoker	5 (33.3%)
HYHA type I	9 (60%)
HYHA type II	4 (26.7%)
HYHA type III	2 (13.3%)
HYHA type IV	1 (6.7)
Beta-blocker	9 (60%)
Loop diuretic	3 (20%)
Other diuretic	1 (6.7%)
Ca channel blocker	1 (6.7%)
ARB blocker	0 (0%)
ACEi	6 (40%)

Abbreviations: ACEi angiotensin-converting enzyme inhibitors, ARBs angiotensin-receptor antagonists, BSA body surface area, Ca calcium, DM diabetes mellitus, HTN hypertension, MI myocardial infarction.

**Table 2 medsci-11-00013-t002:** Descriptive statistics of recorded parameters.

Groups	Mean ± SD
4D flow derived	
Aortic valve backward flow (mL)	1.4 ± 2.6
Mitral valve backward flow (mL)	13.3 ± 13.5
Mitral valve forward flow (mL)	100 ± 28
LV stroke volume (mL)	101 ± 29
AI-derived without including MVP doming volume in LVESV	
LV end-diastolic volume (mL)	200 ± 66
LV end-systolic volume (mL)	77 ± 33
LV stroke volume (mL)	123 ± 48
LV ejection fraction (%)	61 ± 14
MV ejection fraction (%)	11 ±11
Manually refined including MVP doming volume in LVESV	
LV end-diastolic volume (mL)	202 ± 63
LV end-systolic volume (mL)	98 ± 37
LV stroke volume (mL)	105 ± 38
LV ejection fraction (%)	52 ± 11
MV ejection fraction (%)	13 ± 12

Abbreviations: 4D four-dimensional, AI artificial intelligence, LV left ventricle, mL millilitre, MV mitral valve, MVP mitral valve prolapse. LVESV left ventricular end-systolic volume.

## Data Availability

Underlying data: access to the raw images of patients is not permitted since specialised post-processing imaging-based solutions can identify the study patients in the future.

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
