# Peer review of "The Importance of Mitral Valve Prolapse Doming Volume in the Assessment of Left Ventricular Stroke Volume with Cardiac MRI"

_medsci, 2023, doi:10.3390/medsci11010013_

Round 1

Reviewer 1 Report

In this study, Li et.al. proposed a more precise method to evaluate the LV volume and function for MVP patients by considering the MVP LV doming volume. Authors compared the LV stroke volume with or without the MVP LV doming volume and used 4D flow as the reference.

Overall, this study has a good rationale and clear hypothesis, detailed method section and appropriate references. However, it can be further improved by addressing the comments below:

Comments: 

1.     From introduction section, if 4DF assessment is already providing a very accurate measurement for LV SV, what is the motivation of testing your method and further incorporate it into future clinical practice?

2.     In method section, line96, how does manual correction is performed?

3.     Figure 1: 

a.     Figure 1 should be discussed in result section rather than material and method section. 

b.     Figure legend is not clear representing the different panels in figure. Eg. What does the green line indicate?

c.     Figure 1e&f showed the contour area after manual refinement, what is automatically AI contour looks like? A before and after comparison is missing.

d.     Figure legend needs further text edit.

4.     Since MVP are significantly affecting female population, did you perform your analysis within same gender, is there any difference between male and female patients?

5.     In limitation section, author mainly discussed the limitation of the property of current study, sample size and statical analysis, other limitations of current study should also be discussed, for instance: gender, age or pre-existing conditions of patients were not considered in current study.

6.     In discussion section, clinical implication of current study should be discussed in detail. 

Author Response

Please see the attached revised manuscript.

1. From introduction section, if 4DF assessment is already providing a very accurate measurement for LV SV, what is the motivation of testing your method and further incorporate it into future clinical practice?

Author reply: We thank the insightful reviewer for this comment. 4DF MRI is already a gold standard imaging modality for LVSV quantification. In this paper we used 4DF MRI as a method for further optimisation of LVSV by the most standard methods of short-axis cine stack segmentation which is routinely used across the globe in clinical practice. This is particularly important in patients with MVP where the precision of LVSV can be suboptimal using standard methods as demonstrated in our paper. By factoring in the doming volume this can be improved which will reflect in a better quantification of mitral regurgitation. Therefore the motivation of this paper is to improve the routine practice of left ventricle segmentation of short-axis cine stack. We hope this reassures the insightful reviewer.

2. In method section, line96, how does manual correction is performed?

Author reply: We thank the insightful reviewer this comment. Manual correction is performed to factor in MVP doming volume. We have included ‘Figure 1 penal e&f’ in line 104 to make this clearer.

3. Figure 1: 

a.              Figure 1 should be discussed in result section rather than material and method section. 

b.              Figure legend is not clear representing the different panels in figure. Eg. What does the green line indicate?

c.              Figure 1e&f showed the contour area after manual refinement, what is automatically AI contour looks like? A before and after comparison is missing.

d.              Figure legend needs further text edit.

Author reply: We thank the insightful reviewer this comment. Figure 1 has been revised to include in-detail legends. For panel d&f, AI missed MVP doming volume and therefore there is no contour.

4. Since MVP are significantly affecting female population, did you perform your analysis within same gender, is there any difference between male and female patients?

Author reply: We thank the insightful reviewer this comment. We have a mixed male and female population in this study with majority of female participants (73.3%). This is in keeping with the prevalence of MVP as suggested by the insightful reviewer. 

We have conducted paired t student test with three set of LVSV in male and female gender separately. There is significant difference tested between LVSV4DF and LVSVstandard with p values of 0.04 in female and 0.01 in male group. Same test identified statistical difference between LVSVstandard and LVSVMVP with p value of 0.003 and 0.04.

The ICC results showed a good agreement between LVSV4DF and LVSVstandard with ICC score of 0.76 in female group (p=0.02) and excellent agreement in male group with ICC score of 0.99 (p<0.01). An increased ICC agreement was shown in comparison between LVSV4DF and LVSVMVP with scores of 0.81 in female (p<0.01) and 0.99 in male (p<0.01).

However due to the sample size of each gender group is relatively small (11 female and 4 male), which could lead to an insufficient data power, we have not included this in the manuscript. We hope this reassures the insightful reviewer.

5. In limitation section, author mainly discussed the limitation of the property of current study, sample size and statical analysis, other limitations of current study should also be discussed, for instance: gender, age or pre-existing conditions of patients were not considered in current study.

Author reply: We thank the insightful reviewer for this comment. The limitation of gender and MR severity for our study cohort has been discussed in the discussion section, line 206-213. We hope this reassures the insightful reviewer.

6. In discussion section, clinical implication of current study should be discussed in detail. 

Author reply: We thank the insightful reviewer for this comment. The last paragraph of discussion has been revised to discuss clinical implication in detail. We hope this reassures the insightful reviewer.

Reviewer 2 Report

I have read with great interest the manuscript entitled “The importance of Mitral Valve Prolapse Doming Volume in the assessment of Left Ventricular Stroke Volume with Cardiac MRI” by Rui Li et al submitted as Original Article in Medical Sciences. Briefly, this study aimed to compare LV volumes during end-systolic phases, with and without inclusion of the volume of blood on the left atrial aspect of the atrioventricular groove but still within the MV prolapsing leaflets, against the reference LV SV by four-dimensional flow (4DF). I would like to congratulate the authors for conducting such a study. Nevertheless, several issues should be concerned prior any other evaluation.

1.      Line 45: The authors should refer that the fatal arrhythmic events complicating MR and MVP are a major reason for morbidity and mortality.

2.      Besides CMR usefulness in volumes’ estimation, the authors should add that CMR is useful for detecting MAD and fibrosis, which is associated with arrhythmogenesis.

3.      Line 67 – 70: The authors should add if the patients were consecutively enrolled.

4.      Material and Methods: the authors should state how many operators interpretated the findings.

5.      Results: The authors should add if the MR was primary or secondary and to describe its severity.

6.      Line 138 – 140: The authors should restate this sentence in order to make it more comprehensible.

7.      The authors support that their findings may be useful for the better evaluation of MR severity. It should be stated that a different therapeutic approach is followed in moderate and in severe MR.

8.      Discussion: Moreover, they authors should add if CMR is recommended for MR and MVP according to European and American guidelines.

9. MR and MVP is currently associated with mitral annular disjunction. Its clinical significance for patients’ prognosis should be discussed shorty.

Author Response

 Please see the attached revised manuscript.

Comments:

  1. Line 45: The authors should refer that the fatal arrhythmic events complicating MR and MVP are a major reason for morbidity and mortality.

Author reply: We thank the insightful reviewer for this comment. This has been revised to include the severe arrhythmia association in MVP patients associate with notably higher mortality and mortality. We hope this reassures the insightful reviewer.

  1. Besides CMR usefulness in volumes’ estimation, the authors should add that CMR is useful for detecting MAD and fibrosis, which is associated with arrhythmogenesis.

Author reply: We thank the insightful reviewer for this comment. This has been added to the introduction section (line 51-53).

  1. Line 67 – 70: The authors should add if the patients were consecutively enrolled.

Author reply: We thank the insightful reviewer for this comment. These patients were subslected from a prospective CMR registry with specification inclusion criteria with patients with MVP. This was stated in line 70-71. We hope this reassures the insightful reviewer.

  1. Material and Methods: the authors should state how many operators interpretated the findings.

Author reply: We thank the insightful reviewer. Two experts with more than 10 years of CMR expertise supervised and check the quality of the manual contour, and independently interpret the results of LV volume assessment and 4DF assessment separately. This information has been added to the materials and methods section. We hope this reassures the insightful reviewer.

  1. Results: The authors should add if the MR was primary or secondary and to describe its severity.

Author reply: We thank the insightful reviewer. The aetiology of MR was mainly primary in majority of the cases of MV prolapse. MR severity assessment has been added to the materials and methods section and MR severity results have been added to the results section. Table 2 has been updated accordingly. We hope this reassures the insightful reviewer.

  1. Line 138 – 140: The authors should restate this sentence in order to make it more comprehensible.

Author reply: We thank the insightful reviewer for this comment. We have rephrased the sentence to ‘In this study, we highlight the importance of including the MPV left ventricular doming volume within the left ventricular during end-systolic to precisely calculate the LV SV. The MVP volume is the LV volume which lies on the ventricular side of the mitral valve, but on the atrial aspect of the atrioventricular groove during end-systole.’ We hope this reassures the insightful reviewer.

  1. The authors support that their findings may be useful for the better evaluation of MR severity. It should be stated that a different therapeutic approach is followed in moderate and in severe MR.

Author reply: We thank the insightful reviewer for this comment. We have stated this in the discussion section, line 183-185.

  1. Discussion: Moreover, they authors should add if CMR is recommended for MR and MVP according to European and American guidelines.

Author reply: We thank the insightful reviewer for this comment. TTE is currently advised for MR intervention, however CMR is recommended under certain conditions. We have added this information in the discussion section, line 167-170. We hope this reassures the insightful reviewer.

  1. MR and MVP is currently associated with mitral annular disjunction. Its clinical significance for patients’ prognosis should be discussed shorty.

Author reply: We thank the insightful reviewer for this comment. The discussion of MAD association with MVP was added in the discussion section, line 186-192. We hope this reassures the insightful reviewer.

Reviewer 3 Report

The authors carried out this study which compared LV volumes during end-systolic phases, with and without inclusion of the volume of blood on the left atrial aspect of the atrioventricular groove but still within the MV prolapsing leaflets, against the reference LV SV by four-dimensional flow (4DF). Fifteen patients with MV prolapse (MVP) were retrospectively enrolled in this study. The authors compared LV SV with (LV SVMVP) and without (LV SVstandard) MVP left ventricular doming volume, using 4D flow (LV SV4DF) as the reference value. Significant differences were observed when comparing LV SVstandard and LV SVMVP (p<0.001), and between LV SVstandard and LV SV4DF (p=0.02). The Intraclass Correlation Coefficient (ICC) test demonstrated good repeatability between LV SVMVP and LV SV4DF (ICC=0.86, 32 p<0.001) but only moderate repeatability between LV SVstandard and LV SV4DF (ICC=0.75, p<0.01). Calculating LV SV by including the MVP left ventricular doming volume has a higher consistency with LV SV derived from 4DF assessment. The authors  suggest that MVP doming volume should be included to precisely characterise LV volume and function, and the physiology of MVP. 

1. This is overall an informative study albeit with a small number of patients. The authors have stated so in the limitations section. Larger studies are needed to validate the results of this study.

Author Response

Please see attached the revised manuscript.

1. This is overall an informative study albeit with a small number of patients. The authors have stated so in the limitations section. Larger studies are needed to validate the results of this study.

We thank the insightful reviewer for this comment and the encouragement about this paper. We agree that the findings of this study need to be validated in a larger cohort.

Round 2

Reviewer 2 Report

Thank you for incorporating my suggestions. 

Author Response

We thank the insightful reviewer for the comments and encouragement about this paper.